# Automatic Discovery of Cognitive Skills to Improve the Prediction of Student Learning

**Robert V. Lindsey, Mohammad Khajah, Michael C. Mozer**
Department of Computer Science and Institute of Cognitive Science
University of Colorado, Boulder

## Abstract

To master a discipline such as algebra or physics, students must acquire a set of cognitive skills. Traditionally, educators and domain experts use intuition to determine what these skills are and then select practice exercises to hone a particular skill. We propose a technique that uses student performance data to automatically discover the skills needed in a discipline. The technique assigns a latent skill to each exercise such that a student's expected accuracy on a sequence of same-skill exercises improves monotonically with practice. Rather than discarding the skills identified by experts, our technique incorporates a nonparametric prior over the exercise-skill assignments that is based on the expert-provided skills and a weighted Chinese restaurant process. We test our technique on datasets from five different intelligent tutoring systems designed for students ranging in age from middle school through college. We obtain two surprising results. First, in three of the five datasets, the skills inferred by our technique support significantly improved predictions of student performance over the expert-provided skills. Second, the expert-provided skills have little value: our technique predicts student performance nearly as well when it ignores the domain expertise as when it attempts to leverage it. We discuss explanations for these surprising results and also the relationship of our skill-discovery technique to alternative approaches.

## 1 Introduction

With the advent of massively open online courses (MOOCs) and online learning platforms such as Khan Academy and Reasoning Mind, large volumes of data are collected from students as they solve exercises, acquire cognitive skills, and achieve a conceptual understanding. A student's data provides clues as to his or her *knowledge state*—the specific facts, concepts, and operations that the student has mastered, as well as the depth and robustness of the mastery. Knowledge state is dynamic and evolves as the student learns and forgets.

Tracking a student's time-varying knowledge state is essential to an intelligent tutoring system. Knowledge state pinpoints the student's strengths and deficiencies and helps determine what material the student would most benefit from studying or practicing. In short, efficient and effective personalized instruction requires inference of knowledge state [20, 25].

Knowledge state can be decomposed into atomic elements, often referred to as *knowledge components* [7, 13], though we prefer the term *skills*. Skills include retrieval of specific facts, e.g., the translation of 'dog' into Spanish is *perro*, as well as operators and rules in a domain, e.g., dividing each side of an algebraic equation by a constant to transform $3(x + 2) = 15$ into $x + 2 = 5$, or calculating the area of a circle with radius $r$ by applying the formula

$\pi r^2$. When an exercise or question is posed, students must apply one or more skills, and the probability of correctly applying a skill is dependent on their knowledge state.

To predict a student's performance on an exercise, we thus must: (1) determine which skill or skills are required to solve the exercise, and (2) infer the student's knowledge state for those skills. With regard to (1), the correspondence between exercises and skills, which we will refer to as an *expert labeling*, has historically been provided by human experts. Automated techniques have been proposed, although they either rely on an expert labeling which they then refine [5] or treat the student knowledge state as static [3]. With regard to (2), various dynamical latent state models have been suggested to infer time-varying knowledge state given an expert labeling. A popular model, *Bayesian knowledge tracing* assumes that knowledge state is binary—the skill is either known or not known [6]. Other models posit that knowledge state is continuous and evolves according to a linear dynamical system [21].

Only recently have methods been suggested that simultaneously address (1) and (2), and which therefore perform *skill discovery*. Nearly all of this work has involved matrix factorization [24, 22, 14]. Consider a student × exercise matrix whose cells indicate whether a student has answered an exercise correctly. Factorization leads to a vector for each student characterizing the degree to which the student has learned each of $N_{\text{skill}}$ skills, and a vector for each exercise characterizing the degree to which that exercise requires each of $N_{\text{skill}}$ skills. Modeling student learning presents a particular challenge because of the temporal dimension: students' skills improve as they practice. Time has been addressed either via dynamical models of knowledge state or by extending the matrix into a tensor whose third dimension represents time.

We present an approach to skill discovery that differs from matrix factorization approaches in three respects. First, rather than ignoring expert labeling, we adopt a Bayesian formulation in which the expert labels are incorporated into the prior. Second, we explore a nonparametric approach in which the number of skills is determined from the data. Third, rather than allowing an exercise to depend on multiple skills and to varying degrees, we make a stronger assumption that each exercise depends on exactly one skill in an all-or-none fashion. With this assumption, skill discovery is equivalent to the partitioning of exercises into disjoint sets. Although this strong assumption is likely to be a simplification of reality, it serves to restrict the model's degrees of freedom compared to factorization approaches in which each student and exercise is assigned an $N_{\text{skill}}$-dimensional vector. Despite the application of sparsity and nonnegativity constraints, the best models produced by matrix factorization have had low-dimensional skill spaces, specifically, $N_{\text{skill}} \leq 5$ [22, 14]. We conjecture that the low dimensionality is not due to the domains being modeled requiring at most 5 skills, but rather to overfitting for $N_{\text{skill}} > 5$. With our approach of partitioning exercises into disjoint skill sets, we can afford $N_{\text{skill}} \gg 5$ without giving the model undue flexibility. We are aware of one recent approach to skill discovery [8, 9] which shares our assumption that each exercise depends on a single skill. However, it differs from our approach in that it does not try to exploit expert labels and presumes a fixed number of skills. We contrast our work to various alternative approaches toward the end of this paper.

## 2 A nonparametric model for automatic skill discovery

We now introduce a generative probabilistic model of student problem-solving in terms of two components: (1) a prior over the assignment of exercises to skills, and (2) the likelihood of a sequence of responses produced by a student on exercises requiring a common skill.

### 2.1 Weighted CRP: A prior on skill assignments

Any instructional domain (e.g., algebra, geometry, physics) has an associated set of exercises which students must practice to attain domain proficiency. We are interested in the common situation where an expert has identified, for each exercise, a specific skill which is required for its solution (the expert labeling). It may seem unrealistic to suppose that each exercise requires no more than one skill, but in intelligent tutoring systems [7, 13], complex exercises (e.g., algebra word problems) are often broken down into a series of steps which are small

enough that they could plausibly require only one skill (e.g., adding a constant to both sides of an algebraic equation). Thus, when we use the term 'exercise', in some domains we are actually referring to a step of a compound exercise. In other domains (e.g., elementary mathematics instruction), the exercises are designed specifically to tap what is being taught in a lesson and are thus narrowly focused.

We wish to exploit the expert labeling to design a nonparametric prior over assignments of exercises to skills—hereafter, *skill assignments*—and we wish to vary the strength of the *bias* imposed by the expert labeling. With a strong bias, the prior would assign nonzero probability to only the expert labeling. With no bias, the expert labeling would be no more likely than any other. With an intermediate bias, which provides soft constraints on the skill assignment, a suitable model might improve on the expert labeling.

We considered various methods, including fragmentation-coagulation processes [23] and the distance-dependent Chinese restaurant process [4]. In this article, we describe a straightforward approach based on the Chinese restaurant process (CRP) [1], which induces a distribution over partitions. The CRP is cast metaphorically in terms of a Chinese restaurant in which each entering customer chooses a table at which to sit. Denoting the table at which customer $i$ sits as $Y_i$, customer $i$ can take a seat at an occupied table $y$ with $P(Y_i = y) \propto n_y$ or at an empty table with $P(Y_i = N_{\text{table}} + 1) \propto \alpha$, where $N_{\text{table}}$ is the number of occupied tables and $n_y$ is the number of customers currently seated at table $y$.

The *weighted Chinese restaurant process* (*WCRP*) [10] extends this metaphor by supposing that customers each have a fixed *affiliation* and are biased to sit at tables with other customers having similar affiliations. The WCRP is nothing more than the posterior over table assignments given a CRP prior and a likelihood function based on affiliations. In the mapping of the WCRP to our domain, customers correspond to exercises, tables to distinct skills, and affiliations to expert labels. The WCRP thus partitions the exercises into groups sharing a common skill, with a bias to assign the same skill to exercises having the same expert label.

The WCRP is specified in terms of a set of parameters $\boldsymbol{\theta} \equiv \{\theta_1, \ldots, \theta_{N_{\text{table}}}\}$, where $\theta_y$ represents the affiliation associated with table $y$. In our domain, the affiliation corresponds to one of the expert labels: $\theta_y \in \{1, \ldots, N_{\text{skill}}\}$. From a generative modeling perspective, the affiliation of a table influences the affiliations of each customer seated at the table. Using $X_i$ to denote the affiliation of customer $i$—or equivalently, the expert label associated with exercise $i$—we make the generative assumption:

$$P(X_i = x | Y_i = y, \boldsymbol{\theta}) \propto \beta \delta_{x, \theta_y} + 1 - \beta ,$$

where $\delta$ is the Kronecker delta and $\beta$ is the previously mentioned bias. With $\beta = 0$, a customer is equally likely to have any affiliation; with $\beta = 1$, all customers at a table will have the table's affiliation. With uniform priors on $\theta_y$, the conditional distribution on $\theta_y$ is:

$$P(\theta_y | \mathbf{X}^{(y)}) \propto (1 - \beta)^{-n_y^{\theta_y}}$$

where $\mathbf{X}^{(y)}$ is the set of affiliations of customers seated at table $y$ and $n_y^a \equiv \sum_{X_i \in \mathbf{X}^{(y)}} \delta_{x_i, a}$ is the number of customers at table $y$ with affiliation $a$.

Marginalizing over $\boldsymbol{\theta}$, the WCRP specifies a distribution over table assignments for a new customer: an occupied table $y \in \{1, \ldots, N_{\text{table}}\}$ is chosen with probability

$$P(Y_i = y | X_i, \mathbf{X}^{(y)}) \propto n_y \; \frac{1 + \beta(\kappa_y^{x_i} - 1)}{1 + \beta(N_{\text{skill}}^{-1} - 1)}, \quad \text{with} \quad \kappa_y^a \equiv \frac{(1 - \beta)^{-n_y^a}}{\sum_{\tilde{a}=1}^{N_{\text{skill}}} (1 - \beta)^{-n_y^{\tilde{a}}}}. \quad (1)$$

$\kappa_y^a$ is a softmax function that tends toward 1 if $a$ is the most common affiliation among customers at table $y$, and tends toward 0 otherwise. In the WCRP, an empty table $N_{\text{table}} + 1$ is selected with probability

$$P(Y_i = N_{\text{table}} + 1) \propto \alpha. \quad (2)$$

We choose to treat $\alpha$ not as a constant but rather define $\alpha \equiv \alpha'(1 - \beta)$ where $\alpha'$ becomes the free parameter of the model that modulates the expected number of occupied tables, and the term $1 - \beta$ serves to give the model less freedom to assign new tables when the

affiliation bias is high. (We leave the constant in the denominator of Equation 1 so that $\alpha$ has the same interpretation regardless of $\beta$.)

For $\beta = 0$, the WCRP reduces to the CRP and expert labels are ignored. Although the WCRP is undefined for $\beta = 1$, it is defined in the limit $\beta \to 1$, and it produces a seating arrangement equivalent to the expert labels with probability 1. For intermediate $\beta$, the expert labels serve as an intermediate constraint. For any $\beta$, the WCRP seating arrangement specifies a skill assignment over exercises.

## 2.2    BKT: A theory of human skill acquisition

In the previous section, we described a prior over skill assignments. Given an assignment, we turn to a theory of the temporal dynamics of human skill acquisition. Suppose that a particular student practices a series of exercises, $\{e_1, e_2, \ldots, e_t, \ldots, e_T\}$, where the subscript indicates order and each exercise $e_t$ depends on a corresponding skill, $s_t$.[1] We assume that whether or not a student responds correctly to exercise $e_t$ depends solely on the student's mastery of $s_t$. We further assume that when a student works on $e_t$, it has no effect on the student's mastery of other skills $\tilde{s}$, $\tilde{s} \neq s_t$. These assumptions—adopted by nearly all past models of student learning—allow us to consider each skill independently of the others. Thus, for skill $\tilde{s}$, we can select its subset of exercises from the sequence, $\mathbf{e}^{\tilde{s}} = \{e_t \mid s_t = \tilde{s}\}$, preserving order in the sequence, and predict whether the student will answer each exercise correctly or incorrectly. Given the uncertainty in such predictions, models typically predict the joint likelihood over the sequence of responses, $P(R_1, \ldots, R_{|\mathbf{e}^{\tilde{s}}|})$, where the binary random variable $R_t$ indicates the correctness of the response to $e_t$.

The focus of our research is not on developing novel models of skill acquisition. Instead, we incorporate a simple model that is a mainstay of the field, Bayesian knowledge tracing (BKT) [6]. BKT is based on a theory of all-or-none human learning [2] which postulates that a student's knowledge state following trial $t$, $K_t$, is binary: 1 if the skill has been mastered, 0 otherwise. BKT is a hidden Markov model (HMM) with internal state $K_t$ and emissions $R_t$.

Because BKT is typically used to model practice over brief intervals, the model assumes no forgetting, i.e., $K$ cannot transition from 1 to 0. This assumption constrains the time-varying knowledge state: it can make at most one transition from 0 to 1 over the sequence of trials. Consequently, the $\{K_t\}$ can be replaced by a single latent variable, $T$, that denotes the trial following which a transition is made, leading to the BKT generative model:

$$P(T = t | \lambda_L, \lambda_M) = \begin{cases} \lambda_L & \text{if } t = 0 \\ (1 - \lambda_L)\lambda_M(1 - \lambda_M)^{t-1} & \text{if } t > 0 \end{cases} \tag{3}$$

$$P(R_t = 1 | \lambda_G, \lambda_S, T) = \begin{cases} \lambda_G & \text{if } i \leq T \\ 1 - \lambda_S & \text{otherwise,} \end{cases} \tag{4}$$

where $\lambda_L$ is the probability that a student has mastered the skill prior to performing the first exercise, $\lambda_M$ is the transition probability from the not-mastered to mastered state, $\lambda_G$ is the probability of correctly *guessing* the answer prior to skill mastery, and $\lambda_S$ is the probability of answering incorrectly due to a *slip* following skill mastery.

Although we have chosen to model student learning with BKT, any other probabilistic model of student learning could be used in conjunction with our approach to skill discovery, including more sophisticated variants of BKT [11] or models of knowledge state with continuous dynamics [21]. Further, our approach does not require BKT's assumption that learning a skill is conditionally independent of the practice history of other skills. However, the simplicity of BKT allows one to conduct modeling on a relatively large scale.

# 3 Implementation

We perform posterior inference through Markov chain Monte Carlo (MCMC) sampling. The conditional probability for $Y_i$ given the other variables is proportional to the product of the WCRP prior term and the likelihood of each student's response sequence. The prior term is given by Equations 1 and 2, where by exchangeability we can take $Y_i$ to be the last customer to enter the restaurant and where we analytically marginalize $\boldsymbol{\theta}$. For an existing table, the likelihood is given by the BKT HMM emission sequence probability. For a new table, we must add an extra step to calculating the emission sequence probability because the BKT parameters do not have conjugate priors. We used Algorithm 8 from [16], which effectively produces a Monte Carlo approximation to the intractable marginal data likelihood, integrating out over the BKT parameters that could be drawn for the new table.

For lack of conjugacy and any strong prior knowledge, we give each table's $\lambda_L$, $\lambda_M$, and $\lambda_S$ independent uniform priors on $[0, 1]$. Because we wish to interpret BKT's $K = 1$ state as a "learned" state, we parameterize $\lambda_G$ as being a fraction of $1 - \lambda_S$, where the fraction has a uniform prior on $[0, 1]$. We give $\log(1 - \beta)$ a uniform prior on $[-5, 0]$ based on the simulations described in Section 4.1, and $\alpha'$ is given an improper uniform prior with support on $\alpha' > 0$. Because of the lack of conjugacy, we explicitly represent each table's BKT parameters during sampling. In each iteration of the sampler, we update the table assignments of each exercise and then apply five axis-aligned slice sampling updates to each table's BKT parameters and to the hyperparameters $\beta$ and $\alpha'$ [17].

For all simulations, we run the sampler for 200 iterations and discard the first 100 as the burn-in period. The seating arrangement is initialized to the expert-provided skills; all other parameters are initialized by sampling from the generative model. We use the post burn-in samples to estimate the expected posterior probability of a student correctly responding in a trial, integrating out over uncertainty in all skill assignments, BKT parameterizations, and hyperparameters. We explored using more iterations and a longer burn-in period but found that doing so did not yield appreciable increases in training or test data likelihoods.

# 4 Simulations

## 4.1 Sampling from the WCRP

We generated synthetic exercise-skill assignments via a draw from a CRP prior with $\alpha = 3$ and $N_{\text{exercise}} = 100$. Using these assignments as both the ground-truth and expert labels, we then simulated draws from the WCRP to determine the effect of $\beta$ (the expert labeling bias) and $\alpha'$ (concentration scaling parameter; see Equation 2) on the model's behavior. Figure 1a shows the *reconstruction score*, a measure of similarity between the induced assignment and the true labels. This score is the difference between (1) the proportion of pairs of exercises that belong to the same true skill that are assigned to the same recovered skill, and (2) the proportion of pairs of exercises that belong to different true skills that are assigned to different recovered skills. The score is in $[0, 1]$, with 0 indicating no better than a chance relationship to the true labels, and 1 indicating the true labels are recovered exactly. The reported score is the mean over replications of the simulation and MCMC samples. As $\beta$ increases, the recovered skills better approximate the expert (true) skills, independent of $\alpha'$. Figure 1b shows the expected interaction between $\alpha'$ and $\beta$ on the number of occupied tables (induced skills): only when the bias is weak does $\alpha'$ have an effect.

## 4.2 Skill recovery from synthetic student data

We generated data for $N_{\text{student}}$ synthetic students responding to $N_{\text{exercise}}$ exercises presented in a random order for each student. Using a draw from the CRP prior with $\alpha = 3$, we generated exercise-skill assignments. For each skill, we generated sequences of student correct/incorrect responses via BKT, with parameters sampled from plausible distributions: $\lambda_L \sim \text{Uniform}(0, 1)$, $\lambda_M \sim \text{Beta}(10, 30)$, $\lambda_G \sim \text{Beta}(1, 9)$, and $\lambda_S \sim \text{Beta}(1, 9)$.

Figure 1c shows the model's reconstruction of true skills for 24 replications of the simulation with $N_{\text{student}} = 100$ and $N_{\text{exercise}} = 200$, varying $\beta$, providing a set of expert skill labels that were either the *true* labels or a *permutation* of the true labels. The latter conveys no information about the true labels. The most striking feature of the result is that the model

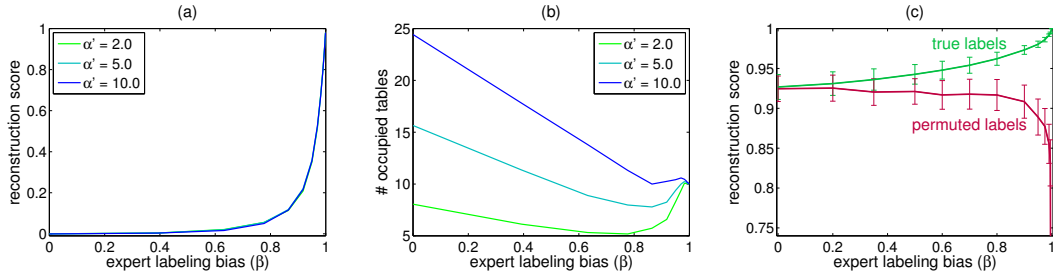

Figure 1: (a,b) Effect of varying expert labeling bias ($\beta$) and $\alpha'$ on sampled skill assignments from a WCRP; (c) Effect of expert labels and $\beta$ on the full model's reconstruction of the true skills from synthetic data

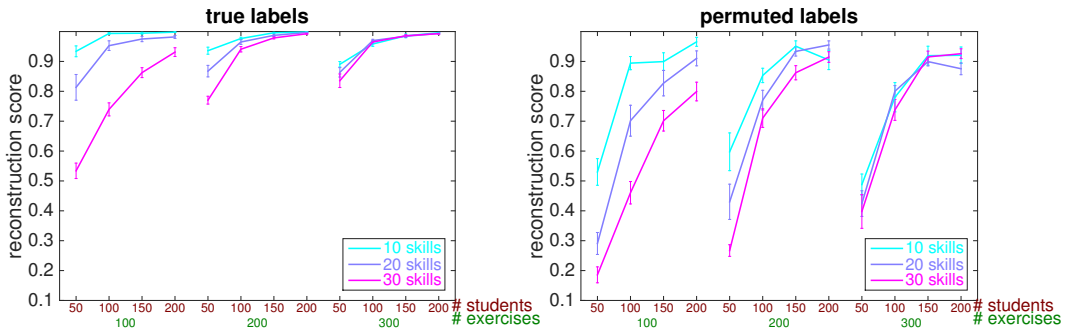

Figure 2: Effect of expert labels, $N_{\text{student}}$, $N_{\text{exercise}}$, and $N_{\text{skill}}$ on the model's reconstruction of the true skills from synthetic data

does an outstanding job of reconstructing the true labeling whether the expert labels are correct or not. Only when the bias $\beta$ is strong and the expert labels are erroneous does the model's reconstruction performance falter. The bottom line is that a good expert labeling can help, whereas a bad expert labeling should be no worse than no expert-provided labels.

In a larger simulation, we systematically varied $N_{\text{student}} \in \{50, 100, 150, 200\}$, $N_{\text{exercise}} \in \{100, 200, 300\}$, and assigned the exercises to one of $N_{\text{skill}} \in \{10, 20, 30\}$ skills via uniform multinomial sampling. Figure 2 shows the result from 30 replications of the simulation using expert labels that were either true or permuted (left and right panels, respectively). With a good expert labeling, skill reconstruction is near perfect with $N_{\text{student}} \geq 100$ and an $N_{\text{exercise}} : N_{\text{skill}}$ ratio of at least 10. With a bad expert labeling, more data is required to obtain accurate reconstructions, say, $N_{\text{student}} \geq 200$. As one would expect, a helpful expert labeling can overcome noisy or inadequate data.

### 4.3 Evaluation of student performance data

We ran simulations on five student performance datasets (Table 1). The datasets varied in the number of students, exercises, and expert skill labels; the students in the datasets ranged in age from middle school to college. Each dataset consists of student identifiers, exercise identifiers, trial numbers, and binary indicators of response correctness from students undergoing variable-length sequences of exercises over time.[2] Exercises may appear in different orders for each student and may occur multiple times for a given student.

| source | dataset | # students | # exercises | # trials | # skills (expert) | # skills (WCRP) | $\beta$ (WCRP) |
|---|---|---|---|---|---|---|---|
| PSLC DataShop [12] | fractions game | 51 | 179 | 4,349 | 45 | 7.9 | 0.886 |
| PSLC DataShop [12] | physics tutor | 66 | 4,816 | 110,041 | 652 | 49.4 | 0.947 |
| PSLC DataShop [12] | engineering statics | 333 | 1,223 | 189,297 | 156 | 99.2 | 0.981 |
| [15] | Spanish vocabulary | 182 | 409 | 578,726 | 221 | 183 | 0.996 |
| PSLC DataShop [12] | geometry tutor | 59 | 139 | 5,104 | 18 | 19.7 | 0.997 |

Table 1: Five student performance datasets used in simulations

We compared a set of models which we will describe shortly. For each model, we ran ten replications of five-fold cross validation on each dataset. In each replication, we randomly partitioned the set of all students into five equally sized disjoint subsets. In each replication-fold, we collected posterior samples using our MCMC algorithm given the data recorded for students in four of the five subsets. We then used the samples to predict the response sequences (correct vs. incorrect) of the remaining students. On occasion, students in the test set were given exercises that had not appeared in the training set. In those cases, the model used samples from Equations 1-2 to predict the new exercises' skill assignments.

The models we compare differ in how skills are assigned to exercises. However, every model uses BKT to predict student performance given the skill assignments. Before presenting results from the models, we first need to verify the BKT assumption that students improve on a skill over time. We compared BKT to a baseline model which assumes a stationary probability of a correct response for each skill. Using the expert-provided skills, BKT achieves a mean 11% relative improvement over the baseline model across the five datasets. Thus, BKT with expert-provided skills is sensitive to the temporal dynamics of learning.

To evaluate models, we use BKT to predict the test students' data given the model-specified skill assignment. We calculated several prediction-accuracy metrics, including RMSE and mean log loss. We report area under the ROC curve ($AUC$), though all metrics yield the same pattern of results. Figure 3 shows the mean AUC, where larger AUC values indicate better performance. Each graph is a different dataset. The five colored bars represent alternative approaches to determining the exercise-skill assignments. **LFA** uses skills from Learning Factors Analysis, a semi-automated technique that refines expert-provided skills [5]; LFA skills are available for only the Fractions and Geometry datasets. **Single** assigns the same skill to all exercises. **Exercise specific** assigns a different skill to each exercise. **Expert** uses the expert-provided skills. **WCRP(0)** uses the WCRP with no bias toward the expert-provided skills, i.e., $\beta = 0$, which is equivalent to a CRP. **WCRP($\beta$)** is our technique with the level of bias inferred from the data.

The performance of **expert** is unimpressive. On Fractions, **expert** is worse than the **single** baseline. On Physics and Statics, **expert** is worse than the **exercise-specific** baseline. **WCRP($\beta$)** is consistently better than both the **single** and **exercise-specific** baselines across all five datasets. **WCRP($\beta$)** also outperforms **expert** by doing significantly better on three datasets and equivalently on two. Finally, **WCRP($\beta$)** is about the same as **LFA** on Geometry, but substantially better on Fractions. (A comparison between these models is somewhat inappropriate. **LFA** has an advantage because it was developed on Geometry and is provided entire data sets for training, but it has a disadvantage because it was not designed to improve the performance of BKT.) Surprisingly, **WCRP(0)**, which ignores the expert-provided skills, performs nearly as well as **WCRP($\beta$)**. Only for Geometry was **WCRP($\beta$)** reliably better (two-tailed $t$-test with $t(49) = 5.32$, $p < .00001$). The last column of Table 1, which shows the mean inferred $\beta$ value for **WCRP($\beta$)**, helps explain the pattern of results. The datasets are arranged in order of smallest to largest inferred $\beta$, both in Table 1 and Figure 3. The inferred $\beta$ values do a good job of indicating where **WCRP($\beta$)** outperforms **expert**: the model infers that the expert skill assignments are useful for Geometry and Spanish, but less so for the other datasets. Where the expert skill assignments are most useful, **WCRP(0)** suffers. On the datasets where **WCRP($\beta$)** is highly biased, the mean number of inferred skills (Table 1, column 7) closely corresponds to the number of expert-provided skills.

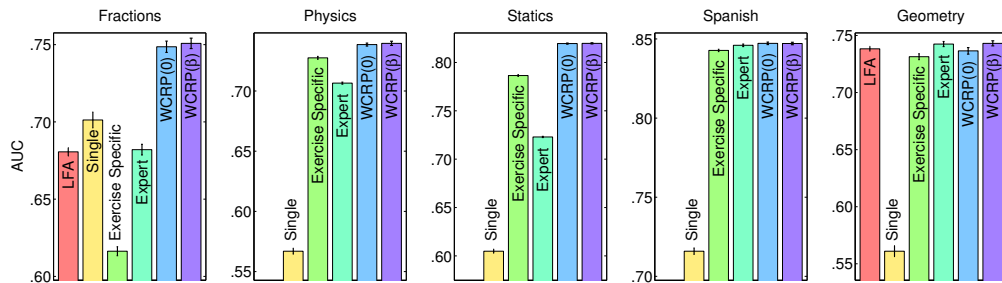

Figure 3: Mean AUC on test students' data for six different methods of determining skill assignments in BKT. Error bars show ±1 standard error of the mean.

## 5    Discussion

We presented a technique that discovers a set of cognitive skills which students use for problem solving in an instructional domain. The technique assumes that when a student works on a sequence of exercises requiring the same skill, the student's expected performance should monotonically improve. Our technique addresses two challenges simultaneously: (1) determining which skill is required to correctly answer each exercise, and (2) modeling a student's dynamical knowledge state for each skill. We conjectured that a technique which jointly addresses these two challenges might lead to more accurate predictions of student performance than a technique which was based on expert skill labels. We found strong evidence for this conjecture: On 3 of 5 datasets, skill discovery yields significantly improved predictions over fixed expert-labeled skills; on the other two datasets, the two approaches obtain comparable results.

Counterintuitively, incorporating expert labels into the prior provided little or no benefit. Although one expects prior knowledge to play a smaller role as datasets become larger, we observed that even medium-sized datasets (relative to the scale of today's big data) are sufficient to support a pure data-driven approach. In simulation studies with both synthetic data and actual student datasets, 50-100 students and roughly 10 exercises/skill provides strong enough constraints on inference that expert labels are not essential.

Why should the expert skill labeling ever be worse than an inferred labeling? After all, educators design exercises to help students develop particular cognitive skills. One explanation is that educators understand the knowledge structure of a domain, but have not parsed the domain at the right level of granularity needed to predict student performance. For example, a set of exercises may all tap the same skill, but some require a deep understanding of the skill whereas others require only a superficial or partial understanding. In such a case, splitting the skill into two subskills may be beneficial. In other cases, combining two skills which are learned jointly may subserve prediction, because the combination results in longer exercise histories which provide more context for prediction. These arguments suggest that fragmentation-coagulation processes [23] may be an interesting approach to leveraging expert labelings as a prior.

One limitation of the results we report is that we have yet to perform extensive comparisons of our technique to others that jointly model the mapping of exercises to skills and the prediction of student knowledge state. Three matrix factorization approaches have been proposed, two of which are as yet unpublished [24, 22, 14]. The most similar work to ours, which also assumes each exercise is mapped to a single skill, is the *topical HMM* [8, 9]. The topical HMM differs from our technique in that the underlying generative model supposes that the exercise-skill mapping is inherently stochastic and thus can change from trial to trial and student to student. (Also, it does not attempt to infer the number of skills or to leverage expert-provided skills.) We have initated collaborations with several authors of these alternative approaches, with the goal of testing the various approaches on exactly the same datasets with the same evaluation metrics.

**Acknowledgments** This research was supported by NSF grants BCS-0339103 and BCS-720375 and by an NSF Graduate Research Fellowship to R. L.

## Footnotes

[1]To tie this notation to the notation of the previous section, $s_t \equiv y_{e_t}$, i.e., the table assignments of the WCRP correspond to skills, and exercise $e_t$ is seated at table $y_{e_t}$. Note that $i$ in the previous section was used as an index over distinct exercises, whereas $t$ in this section is used as an index over trials. The same exercise may be presented multiple times.

[2] For the DataShop datasets, exercises were identified by concatenating what they call the problem hierarchy, problem name, and the step name columns. Expert-provided skill labels were identified by concatenating the problem hierarchy column with the skill column following the same practice as in [19, 18]. The expert skill labels infrequently associate an exercise with multiple skills. For such exercises, we treat the combination of skills as one unique skill.

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
