[Reviews · NeurIPS 2014]

Submitted by Assigned_Reviewer_29

This paper proposes a probabilistic approach for learning the assignment of exercises to skills from student data, where student knowledge changes while exercises are being solved; the model also estimates the student knowledge while estimating the skill assignments. The paper uses a weighted CRP to model the assignment, incorporating expert labelings through the weighting. In simulation, the method recovers skill labelings with high accuracy, with little dependence on the expert labels, and across several datasets, the paper finds that skill labelings from this method result in higher prediction accuracy than other approaches.

Overall, I found the paper to be clear and the proposed model is a relatively novel extension of existing methods. The evaluation methods demonstrate that the approach is likely to be effective for relevant applications, and that its inferences are accurate. I appreciated the inclusion of both synthetic data and a variety of real datasets. One interesting issue I found in the evaluation is that while the single skill assumption seems partially justified by the fact that these exercises were created to tap a single skill, the expert skills are in many cases a poor fit for prediction, suggesting that the experts may not be clear about the connection between each item and skill. Some discussion of the poor fit of the expert model is included in the discussion, but I think this remains an important point to consider, both in the motivation of the model and in trying to determine what effect using these new skills might have for control policies for tutors. One point that is not touched on is whether ordering/selection effects in terms of what items each student performs and in what order may skew the results; it would be great if a small amount of space in the paper could be devoted to addressing this issue, given that in typical datasets from tutoring systems one cannot count on items being selected randomly or administered in random order.

The main concern I had with the paper was the presentation of alternative methods, specifically in the paragraphs around lines 57-94 of the intro. The paper first claims that automated techniques either "rely on an expert labeling which they then refine or treat the student knowledge as static." However, a later paragraph notes the Gonzalez-Brenes and Mostow work which does not make this assumption (note: I am assuming this is what is meant, rather than the listed reference numbers of 8-9 which are BKT). Additionally, the matrix factorization methods that are mentioned seem to be cases of "treating the student knowledge as static" and thus might be cited in this sentence as well (e.g., Desmarais' work or the SPARFA work). I noticed that the Lan et al. factorization work was included in the references, but could not find it in the main paper; it also seems relevant here and is an example of a model with time-varying student knowledge that does not rely on an expert labeling.
Update: Thanks for the clarification in the rebuttal about the references - I'm leaving the notes here in case it's helpful to make sure things are fixed and because I think the apparent contradiction between the claims about existing methods and the Gonzalez-Brenes and Mostow work should be addressed, but the rebuttal addresses my concerns.

Also in the introduction, the paper notes that the "best models produced by matrix factorization have had low-dimensional skill spaces, specifically N_skill = 2"; it would be helpful to provide a citation here to clarify the context, given that this is not true for a number of results reported in papers about creating Q-matrices (e.g., Tatsuoka's original fraction work, Desmarais's refinement of these Q-matrices).

Minor note:
On line 425, the same issue with reference numbers occurs as in the intro (8-9 are substituted when I suspect 10-11 are intended).
Summary: The work presented in this paper is novel and clearly carried out, with appropriate evaluation and model design.

Submitted by Assigned_Reviewer_30

The paper describes a system for clustering educational exercises (e.g., math problems) so that exercises requiring the same skill are assigned to the same cluster (and exercises requiring different skills are assigned to different clusters). Clustering is performed using a weighted Chinese Restaurant Process (wCRP). (The weights are used in the prior probability distribution. They come from experts who evaluate the skills that are needed to successfully complete each exercise.) The wCRP is then coupled to an HMM (developed by other authors in earlier publications) for the purpose of predicting whether a person will respond correctly on an exercise (based on the sequence of exercises the student has attempted in the past).

Caveat: I know nothing about the field of education. This paper might be a weak or strong contribution to that field, but I am the wrong person to judge. Consequently, this is a difficult paper for me to review.

Keeping in mind the caveat above, my sense is that the proposed model is a thoughtful, perhaps clever application of machine learning methodologies to the field of education. It did not seem obvious to me that clustering would be a useful methodological approach to the problem studied here, but this paper demonstrates that my intuitions are wrong. Moreover, the simulation results reported in the manuscript seem impressive.
Summary: My sense is that the proposed model is a thoughtful, perhaps clever application of machine learning methodologies to the field of education. However, I do not have the background to properly evaluate this submission.

Submitted by Assigned_Reviewer_41

This paper proposes an approach to automatic discovery of cognitive skills from student performance data, and incorporates a non-parametric prior to identify new skills after starting with an initial set of expert provided skills. The authors show empirically that the learning process results in significant improvement over the set of expert provided skills, which surprisingly have (very) limited value.

Quality: The paper is technically sound, and the authors provide experimental validation of their claims.

Clarity: Well-written, and easy to follow.

Originality: In my opinion, this area is where the main weakness of this paper lies. The research described in this paper is well-motivated and formulated, but the main novelties in this paper lie in applying existing techniques to the task of analysis of cognitive skills. While the approach is reasonable, and some of the experimental findings surprising (and it would be interesting to see a deeper, data-driven analysis of why the expert provided information is so weak, and how that could be improved), I did not find many interesting takeaway messages for a community that wasn't strongly focussed on the application area of intelligent tutoring analysis.

Significance: The paper presents a novel approach to the task of discovering cognitive skills from student performance data. The system is capable of starting with a set of skill labels provided by an expert, and expanding this set from the data. The authors present some surprising analysis of results regarding the effectiveness of expert skill labels that should be worth investigating in greater detail for researchers working in this area. However, there are limited conclusions of general interest to the NIPS community to be drawn from this paper, which strikes me as a strongly application driven work (albeit interesting work).
Summary: I found the paper to be an interesting read from the perspective of work in the educational data mining/intelligent tutoring areas. However, the technical novelty of this work is limited, as it largely adapts established techniques to this problem, and while some of the experimental findings are very interesting, the takeaway from them for the general NIPS community appears fairly limited to me.
Author Feedback
Author rebuttal: We thank reviewer 29 for her/his familiarity with the literature and close attention to our citations. Major apologies for the screwed-up citation numbers. They occurred because we anonymized our own work and must not have re-run bibtex. The presentation of related methods should make more sense with the correct citation numbers. (We cite Lan and also Sohl-Dickstein for their dynamical state models using matrix factorization for skill discovery.) Your point is well taken concerning the backdoor presentation of Gonzalez-Brenes & Mostow's (G&M's) evolving and most recent work. We have some concern with this work, however. G&M have results on only one real data set, and we requested this data set from G&M to compare models. Unfortunately, there appears to be a problem with the data set (the training and test sets are highly overlapping; G&M have yet to resolve this issue). Consequently, we weren't able to include a comparison to their work in our submission. Lastly, we thank the reviewer for pointing out matrix factorization work that yields skill spaces with dimensionality higher than 2. The most closely related work to ours, Sohl-Dickstein and Lan et al, chose 2 and 3-5 skills, respectively.

Reviewer 41 has rated the originality of our work in terms of its contribution to machine learning theory, i.e., the novelty of the algorithm and architecture apart from the application domain. From this perspective, we agree that our work makes a fairly marginal contribution. However, we believe that originality should be judged with regard to the application domain, educational data mining (EDM). We've attempted to situate our approach to skill discovery with respect to other approaches in the field. To be blunter than we were in the paper, the majority of this work is hacky and unprincipled, and/or hasn't been well evaluated. The principled approaches involve matrix factorization. Our approach is in some respects simpler than matrix factorization approaches---because skills are assigned in an all-or-none fashion to problems---but we argue and demonstrate that the simplicity may make learning more tractable and the results more interpretable.

To reviewers 30 and 41: Research that is considered as competent and business-as-usual in the NIPS community is often quite innovative in the home community of the application, to the point that the research is viewed with puzzlement or skepticism. We are aware of cases historically in computer vision, robotics, and cognitive science where seminal work was presented at NIPS and only then did acceptance grow within the disciplines. We believe this issue to be true in the field of EDM at present. Having just returned from EDM 2014, where the upset of the conference was a paper that compared hierarchical Bayesian inference to maximum likelihood (i.e., NIPS circa 2000), and where confusion ensued over the distinction between logit and logistic functions, we argue that NIPS is essential for facilitating the flow of ideas to the applied disciplines, which otherwise are resistant to the infiltration of alternative approaches. We also believe that presentation of applied work at NIPS often serves as an inspiration to theoreticians to develop more sophisticated and powerful algorithms.

Reviewer 41 has rated the originality of our work in terms of its contribution to machine learning theory, i.e., the novelty of the algorithm and architecture apart from the application domain. From this perspective, we agree that our work makes a fairly marginal contribution. However, we believe that originality should be judged with regard to the application domain, educational data mining (EDM). We've attempted to situate our approach to skill discovery with respect to other approaches in the field. To be blunter than we were in the paper, the majority of this work is hacky and unprincipled, and/or hasn't been well evaluated. The principled approaches involve matrix factorization. Our approach is in some respects simpler than matrix factorization approaches---because skills are assigned in an all-or-none fashion to problems---but we argue and demonstrate that the simplicity may make learning more tractable and the results more interpretable.

To reviewers 30 and 41: Research that is considered as competent and business-as-usual in the NIPS community is often quite innovative in the home community of the application, to the point that the research is viewed with puzzlement or skepticism. We are aware of cases historically in computer vision, robotics, and cognitive science where seminal work was presented at NIPS and only then did acceptance grow within the disciplines. We believe this issue to be true in the field of EDM at present. Having just returned from EDM 2014, where the upset of the conference was a paper that compared hierarchical Bayesian inference to maximum likelihood (i.e., NIPS circa 2000), and where confusion ensued over the distinction between logit and logistic functions, we argue that NIPS is essential for facilitating the flow of ideas to the applied disciplines, which otherwise are resistant to the infiltration of alternative approaches. We also believe that presentation of applied work at NIPS often serves as an inspiration to theoreticians to develop more sophisticated and powerful algorithms.